# Bayesian Policy Distillation via Offline RL for Lightweight and Fast Inference

## Abstract

High-performance deep reinforcement learning faces tremendous challenges when implemented on cost-effective low-end embedded systems due to its heavy computational burden. To address this issue, we propose a policy distillation method called Bayesian Policy Distillation (BPD), which effectively retrains small-sized neural networks through an offline reinforcement learning approach. BPD exploits Bayesian neural networks to distill already designed high-performance policy networks by adopting value optimizing, behavior cloning, and sparsity-inducing strategies. Simulation results reveal that the proposed BPD successfully compresses the policy networks, making them lighter and achieving faster inference time. Furthermore, the proposed approach is demonstrated with a real inverted pendulum system and reduced the inference time and memory size by 78 % and 98 %, respectively.

## 1    Introduction

Recently, deep reinforcement learning (DRL) has achieved super-human performances in complex games (Lample & Chaplot, 2017; Schrittwieser et al., 2020) and provided feasible solutions to challenging feedback control problems without the need for system modeling and parameter tuning (Hwangbo et al., 2017; Gu et al., 2017). However, such unprecedented innovations require high computational costs, large memory storage spaces, and long inference times, which limit their applicability to real systems. The realization of DRL that meets realistically acceptable specifications while maintaining the expected high performance is crucial. Research on compressing deep neural networks for practical applications has been actively pursued, particularly outside DRL. Network pruning is the most basic and classical approach and involves removing uninfluential weight parameters based on certain criteria with minimal performance loss (Reed, 1993; LeCun et al., 1989; Lebedev & Lempitsky, 2016; Molchanov et al., 2017; Zhao et al., 2019). Network quantization is another approach to achieving lightweight compressed neural networks by representing traditional 32- or 64-bit floating point values with lower bit precision (Achterhold et al., 2018; Gil et al., 2021). In addition, knowledge distillation (Gou et al., 2021) is widely used to compress deep neural networks. This method transfers knowledge from a large- to small-sized neural network, which has proven effective across various domains such as computer vision and natural language processing (Luo et al., 2017; Li et al., 2019; Deng et al., 2020).

The above methods have continued to evolve to handle large language models (Wang et al., 2020; Ganesh et al., 2021) and embedded systems with limited computation capacity (Wofk et al., 2019; Chen et al., 2020; Naik et al., 2021). Therefore, grafting the above network compression techniques onto DRL would be meaningful and timely. Unlike open-loop-based supervised and unsupervised learning, DRL operates in a closed-loop manner, making network compression challenging. Consequently, network compression techniques have been relatively underexplored in DRL despite their long-standing importance. A few attempts to compress the agent's policy in DRL have been made within an online reinforcement learning (RL) framework (Tan et al., 2023; Baek et al., 2023), which inherently involves the agent interacting with its environment. Practically, without careful parameter tuning, such a direct online approach may degrade performance or stability (Sokar et al., 2021) because the agent's policy is trained and compressed based on environmental interactions. To remedy this issue, the Teacher–Student framework is commonly used in RL literature (Ross & Bagnell, 2010; Hinton et al., 2015; Rusu et al., 2015). This framework consists of the teacher and student policies, with the teacher policy having a larger neural network size. The teacher policy facili-

tates knowledge transfer to the student policy through supervised learning by minimizing the mean squared-error loss or the KullbackLeibler (KL) divergence between them (Rusu et al., 2015; Jang et al., 2020). However, simply mimicking the "teacher policy" to train the "student policy" makes it challenging to learn an effective policy beyond the teacher policy. In addition, most RL algorithms using the Teacher–Student framework rely on online RL and inherently suffer from the aforementioned drawbacks. Thus, a naively and overly compressed student policy may lead to significantly diminished performance compared to the teacher policy (Qu et al., 2022). This challenge underscores the need for leveraging an open-loop-based learning framework such as *offline* RL that aims to learn an improved policy based on pre-collected data generated from the teacher policy.

To tackle these challenges, we propose an efficient offline policy compression algorithm, namely *Bayesian policy distillation* (BPD). BPD integrates offline RL framework and Bayesian neural network (Lampinen & Vehtari, 2001) to create an extremely sparse policy that is lightweight, fast, and energy-efficient. This approach holds promise, particularly in scenarios where computational performance is limited, such as on-board devices or a single cloud server needing to perform neural network computation for multiple client devices. Additionally, BPD is a fully offline learning method, particularly suitable for domains such as healthcare or robotics, where data collection is expensive and implementing online RL poses challenges.

We evaluated the proposed algorithm on the multi-joint dynamics with contact MUJOCO continuous control benchmark (Todorov et al., 2012), widely used in RL. The results confirm that our proposed algorithm can create an extremely sparse policy while preserving or improving the teacher policy performance. Additionally, experiments on a real inverted pendulum demonstrated the practicality of our proposed algorithm and highlighted its potential for real-world applications.

## 2 PRELIMINARY BACKGROUND

This section briefly introduces the background to facilitate understanding of the proposed algorithm.

### 2.1 STANDARD REINFORCEMENT LEARNING

In the standard RL framework, an environment is modeled as a Markov decision process defined as a tuple $(\mathcal{S}, \mathcal{A}, R, P, \gamma)$, where $\mathcal{S}$ and $\mathcal{A}$ are the state and action spaces, respectively; $R : \mathcal{S} \times \mathcal{A} \mapsto \mathbb{R}$ is the reward function; $P : \mathcal{S} \times \mathcal{A} \times \mathcal{S} \mapsto [0, 1]$ is the transition probability function; and $\gamma \in (0, 1)$ is the discount factor. A stochastic policy $\pi : \mathcal{S} \times \mathcal{A} \mapsto [0, 1]$ maps state-action pairs to probability distributions over $\mathcal{S}$. The objective of the standard RL framework is to find an optimal policy that maximizes the discounted cumulative rewards.

The state and action at time $t$ are denoted by $s_t$ and $a_t$, respectively. For a given $s_t$ and $a_t$, the environment provides an immediate scalar reward $R(s_t, a_t)$, and transitions to a new state $s_{t+1} \in \mathcal{S}$ with probability $p(s_{t+1}|s_t, a_t)$. The objective is to find an optimal policy $\pi^*$ that maximizes the discounted cumulative rewards as follows:

$$\pi^* = \arg \max_{\pi} \mathbb{E}_{\tau_0 \sim \pi}[\sum_{t=0}^{\infty} \gamma^t R(s_t, a_t)],$$

where $\tau_0 = (s_0, a_0, s_1, a_1, \cdots)$ is the trajectory under policy $\pi$. The distribution of the trajectories can be written as

$$p(\tau_0) = \rho(s_0) \prod_{t=0}^{\infty} p(s_{t+1}|s_t, a_t)\pi(a_t|s_t),$$

where $\rho(\cdot)$ is the distribution of the initial state $s_0$.

The expected discounted cumulative reward is represented by the following $Q$-function:

$$Q^{\pi}(s_t, a_t) = \mathbb{E}_{\tau_t \sim \pi}[\sum_{k=0}^{\infty} \gamma^k R(s_{t+k}, a_{t+k})|s_t, a_t],$$

where the trajectory $\tau_t$ is $(s_t, a_t, s_{t+1}, a_{t+1}, \cdots)$. The $Q$-function values ($Q$-values) can be represented as the immediate reward plus the discounted future $Q$-values as follows (Bellman, 1957):

$$Q^{\pi}(s_t, a_t) = \mathcal{T}(Q^{\pi}(s_t, a_t)) \triangleq R(s_t, a_t) + \gamma \mathbb{E}_{s_{t+1} \sim P(\cdot|s_t, a_t), a_{t+1} \sim \pi(\cdot|s_{t+1})}[Q(s_{t+1}, a_{t+1})],$$

where $\mathcal{T}$ is the Bellman operator, whose contraction property ensures the convergence of the Q-values with iterative methods. However, since not all of the transition probability functions $p(\cdot|s_t, a_t)$ are known, or the state space $\mathcal{S}$ is too large in some cases, $Q$-values are often computed using sample-based stochastic approximation methods such as $Q$-learning (Watkins & Dayan, 1992). Thus, an optimal policy is obtained by incrementing the $Q$-values for all state-action pairs.

## 2.2 Bayesian Neural Network and Variational Inference

Bayesian neural networks (BNNs) consider their weight parameters as random variables with stochastic distribution. Such a nondeterministic approach offers advantages such as robustness against overfitting and probabilistic interpretability (Gal et al., 2016). Through BNN learning, the posteriors of its weights $\omega$ are approximated for given observations $X\ Y$. Let the posterior be $p(\omega|X, Y)$. Then, we can decompose the posterior using Bayes' rule as follows:

$$p(\omega|X, Y) = \frac{p(Y|X, \omega)p(\omega)}{p(Y|X)}.$$

To approximate the posterior $p(\omega|X, Y)$, the *variational inference* technique has been widely used, which employs a proposal distribution named variational distribution $q_\phi(\omega)$ parameterized by $\phi$ and then makes it close to the posterior by minimizing the following KL divergence:

$$\begin{aligned}
&\mathrm{D_{KL}}(q_\phi(\omega)||p(\omega|X, Y)) \\
&= \mathbb{E}_{q_\phi(\omega)}[\log \frac{q_\phi(\omega)}{p(\omega)}] - \mathbb{E}_{q_\phi(\omega)}[\log p(Y|\omega, X)] + \log p(Y|X) \\
&= \mathrm{D_{KL}}(q_\phi(\omega)||p(\omega)) - \mathbb{E}_{q_\phi(\omega)}[\log p(Y|\omega, X)] + \mathrm{C} \triangleq -\mathcal{L}_{\mathrm{ELBO}} + \mathrm{C}
\end{aligned} \tag{1}$$

Since $\log p(Y|X)$ in (1), called log evidence, is independent of $\phi$ or is unlearnable, we maximize $\mathcal{L}_{\mathrm{ELBO}}$ to minimize the KL divergence between the variational distribution $q_\phi(\omega)$ and posterior $p(\omega|X, Y)$. $\mathcal{L}_{\mathrm{ELBO}}$ can be numerically computed as follows:

$$\mathcal{L}_{\mathrm{ELBO}} \approx \Sigma_{(x_n, y_n) \in \mathcal{D}} \mathbb{E}_{q_\phi(\omega)}[\log p(y_n|\boldsymbol{f}_\omega(x_n))] - \mathrm{D_{KL}}(q_\phi(\omega)||p(\omega)), \tag{2}$$

where $\mathcal{D} = \{(x_n, y_n)\}_{n=1}^{|\mathcal{D}|}$ is a reusable dataset collected earlier, $\boldsymbol{f}_\omega(x_n)$ is the BNN output for input $x_n \in X$, and $y_n \in Y$ is the corresponding groundtruth. Here, we replace $x_n$ and $y_n$ with a state and an action for behavioral cloning (BC) in RL, respectively. To obtain a lightweight performance-aware network model, $\mathcal{L}_{\mathrm{ELBO}}$ is later reflected in learning processing.

## 2.3 Sparse Variational Dropout

Dropout is a common regularization method employed to prevent neural networks from overfitting. Based on the Bernoulli distribution, each neuron in a neural network is assigned the deletion probability (Srivastava et al., 2014). Assigning Bernoulli distribution to each neuron is shown to be equivalent to applying proper Gaussian noises to the random weights characterized by $\theta_{ij}$ and $\alpha_{ij}$ as follows (Wang & Manning, 2013):

$$\omega_{ij} = \theta_{ij}(1 + \sqrt{\alpha_{ij}}\epsilon_{ij}), \tag{3}$$

where $\epsilon_{ij} \sim \mathcal{N}(0, 1)$ and subscript $ij$ denotes the corresponding element in the weight matrix form. The process (3) of generating noisy weights is called Gaussian dropout. In terms of $\theta_{ij}$ and $\alpha_{ij}$ in (3), the proposal distribution $q_\phi(\omega)$ can be expressed element-wise as follows:

$$q_{\phi_{ij}} = q(\omega_{ij}|\theta_{ij}, \alpha_{ij}) = \mathcal{N}(\theta_{ij}, \alpha_{ij}\theta_{ij}^2), \tag{4}$$

where $\phi_{ij} = (\theta_{ij}, \alpha_{ij})$.

If different weights are assumed to be independent, the posterior and prior are fully factorized, and we have

$$\mathrm{D_{KL}}(q_\phi(\omega)||p(\omega)) = \sum_{ij} \mathrm{D_{KL}}(q_\phi(\omega_{ij})||p(\omega_{ij})) = \sum_{ij} \mathrm{D_{KL}}(q(\omega_{ij}|\theta_{ij}, \alpha_{ij})||p(\omega_{ij})),$$

where $q_{\phi(\omega)}$ and $p(\omega)$ are given by

$$q_\phi(\omega) = \prod_{ij} q_{\phi_{ij}}(\omega_{ij}), \qquad p(\omega) = \prod_{ij} p_{ij}(\omega_{ij}).$$

With the factorized form above and minibatch-based stochastic gradient descent method, $\mathcal{L}_{\mathrm{ELBO}}$ in (2) can be rewritten more specifically as follows:

$$\mathcal{L}_{\mathrm{ELBO}} \approx \frac{|\mathcal{D}|}{M} \Sigma_{m=1}^M \mathbb{E}_{q_\phi(\omega)}[\log p(\tilde{y}_m|\boldsymbol{f}_\omega(\tilde{x}_m))] - \sum_{ij} \mathrm{D}_{\mathrm{KL}}(q(\omega_{ij}|\theta_{ij}, \alpha_{ij})||p(\omega_{ij})), \qquad (5)$$

where $|\mathcal{D}|$ is the total number of the dataset, $M$ is the mini-batch size, and $\{(\tilde{x}_m, \tilde{y}_m)\}_{m=1}^M$ represent the observation in the selected mini-batch. To maximize $\mathcal{L}_{\mathrm{ELBO}}$ (5) with random variable $\omega_{ij}$, reparameterization methods such as local reparameterization (Kingma et al., 2015) or additive noise reparameterization (Molchanov et al., 2017) can be applied.

Furthermore, evidence lower bound objective (ELBO) can induce sparsity in BNNs if the prior $p(w_{ij})$ is set to log-uniform distribution that becomes large around zero as follow (Kingma et al., 2015):

$$p(\log(|\omega_{ij}|)) = \mathrm{const.} \leftrightarrow p(|\omega_{ij}|) \propto \frac{1}{|\omega_{ij}|}. \qquad (6)$$

Consequently, by maximizing $\mathcal{L}_{\mathrm{ELBO}}$ in (5) with the log-uniform prior, we can train a highly compressed BNN. This Bayesian network pruning algorithm, called sparse variational dropout, trains individual dropout rate $\alpha_{ij}$ in (4). The network pruning is achieved by removing weights with large $\alpha_{ij}$, as it does not affect the network output significantly. Typically, the threshold $C_{\mathrm{Threshold}} = 3$ is widely used, which corresponds to a binary dropout rate higher than 0.95 (Molchanov et al., 2017). In this case, when $\log(\alpha_{ij})$ exceeds 3, the corresponding weight $\omega_{ij}$ is pruned (i.e., set to zero).

## 2.4 OFFLINE REINFORCEMENT LEARNING

In standard RL, the agent collects data through numerous interactions with the environment, which can be expensive in fields such as robotics and healthcare. This challenge brings attention to offline RL, which allows agents to learn policies without direct interaction with the environment, relying on the dataset collected by a *behavioral* policy in advance. Since the agent can train the policy without engaging in potentially risky interactions, offline RL could be considerably safer and more cost-effective for learning policy than online RL. However, despite its promising paradigm, challenges still remain to overcome in the policy training domain. One main challenge is the undesirable *extrapolation error* of out-of-distribution (OOD) actions (Fujimoto et al., 2019). The *target policy* to train cannot explore state-action pairs that are not included in the static dataset collected by the behavioral policy. Hence, the state-action visitation distribution of the target policy deviates from that of the behavioral policies, leading to optimistic $Q$-values generating OOD actions. This becomes problematic because if an agent is trained with these overly optimistic $Q$-values, it may select poor actions based on the overestimated ones.

Explicit policy constraint approaches have been often employed to address the OOD problem (Kumar et al., 2019; Fujimoto & Gu, 2021; Fakoor et al., 2021). These approaches include introducing regularizing terms to minimize the difference between the visitation distributions of the target and behavioral policies. These regularization terms are approximated as follows:

$$\mathbb{E}_{(s,a)\in\mathcal{D}}[(\pi(s) - a)^2], \qquad (7)$$

where $\pi$ is the target policy and $(s, a)$ is the state-action pair of the behavioral policy stored in the pre-collected dataset $\mathcal{D}$. Minimizing (7), the target policy is trained to inhibit the selection of poor actions that the behavioral policy would not choose. The algorithm proposed in this study alleviates the OOD issue by incorporating such a regularization term, which will be detailed in the next section.

## 3 BAYESIAN POLICY DISTILLATION

### 3.1 BAYESIAN POLICY CONSTRAINT

To incorporate the distillation method into the offline RL framework, we consider the conventional BC approach that trains the target policy through supervised learning using state-action pairs gener-

ated from the behavior policy. Let the collected state-action pairs be $\mathcal{D} = \{(s, a)_n\}_{n=1}^{|\mathcal{D}|}$, where $\mathcal{D}$ is the static dataset, $n$ indexes the samples, and $|\mathcal{D}|$ is the total number of state-action pairs. Then, the student policy is trained by solving the following minimization problem:

$$\min_\omega \mathbb{E}_{(s,a)\sim\mathcal{D}}[(\pi_\omega(s) - a)^2], \tag{8}$$

where $\pi_\omega$ is the target policy parameterized by $\omega$. In conventional policy distillation methods, the student policy network is reduced by naively reducing the hidden layer size, and is trained using BC. However, when the student policy size is excessively small, maintaining the performance of the teacher policy becomes challenging (Rusu et al., 2015).

To avoid the above priori size selection and facilitate BC as in (8), we designed a BNN-based student policy, which makes the policy extremely sparse while preserving the teacher's performance. First, we consider the weights $\omega$ of the student policy as random variables drawn from the distribution $\mathcal{N}(\omega_{ij}|\theta_{ij}, \alpha_{ij}^2\theta_{ij})$, where $\theta_{ij}$ and $\alpha_{ij}$ are independent and learnable. If $(\tilde{x}_m, \tilde{y}_m)$ in (5) is replaced with a state-action pair $(s, a)$, and the prior distribution of $\omega$ is set to be log-uniform as in (6), an RL version of ELBO is constructed as follows:

$$\mathcal{L}_{\text{RL-ELBO}}(\theta, \alpha) = -\underbrace{\frac{|\mathcal{D}|}{M}\Sigma_{m=1}^M \mathbb{E}_{\omega\sim q(\omega|\theta,\alpha)}[(\pi_\omega(s_m) - a_m)^2]}_{\text{BC term}} - \underbrace{D_{\text{KL}}(q(\omega|\theta,\alpha)\|p(\omega))}_{\text{KL term}}. \tag{9}$$

$\mathcal{L}_{\text{RL-ELBO}}$ in (9) includes a term for BC as in (8). The first BC term helps the target (student) policy mimic the behavioral (teacher) policy by learning from state-action pairs generated by the behavioral policy. The second KL term encourages making the target policy network sparse by pushing the weight distribution mean closer to zero. Thus, minimizing $\mathcal{L}_{\text{RL-ELBO}}$ in (9), the student policy mimics the teacher policy and promotes network sparsity.

## 3.2 REFINING THE POLICY WITH VALUE FUNCTION

Thus far, we have focused on training the target policy to have a distribution similar to BC. While this approach helps the target policy avoid selecting poor actions that could deteriorate performance, training agents to perform well when visiting new states is challenging. Therefore, learning a general behavior that makes good actions for states not included in the static dataset is considered when training the $Q$-function.

The $Q$-function parameterized by $\psi$ is trained by minimizing the following loss function:

$$\mathcal{L}_Q(\psi_i) = \mathbb{E}_{(s,a,s')\sim\mathcal{D}}\left[(Q_{\psi_i}(s, a) - y)^2\right], \text{ where } y = R(s, a) + \gamma \min_{i\in\{1,2\}} Q_{\bar{\psi}_i}^{\text{target}}(s', a'), \tag{10}$$

where $s'$ is the next state from the dataset, $a'$ is the next action drawn from the target policy $\pi_\omega(\cdot|s')$, and $Q_{\bar{\psi}_i}^{\text{target}}$ are two target $Q$-functions (Mnih et al., 2015). The smaller of the two target $Q$-values is chosen for updating to prevent overestimation of $Q$-values (Fujimoto et al., 2018; Haarnoja et al., 2018). $a'$ can be viewed as a combination of the action from the deterministic policy $\bar{\pi}_\omega(s)$ and a random perturbation, where $\bar{\pi}_\omega(s)$ represents the mean value of $\pi_\omega(\cdot|s)$. Therefore, sampling $a'$ from $\pi_\omega(\cdot|s')$ can be interpreted as the implicit version of *target policy smoothing regularization* technique introduced in the TD3 algorithm (Fujimoto et al., 2018).

As the $Q$-function is learned, the policy can be updated in the direction of increasing $Q$-value and $\mathcal{L}_{\text{RL-ELBO}}$:

$$\arg\max_{(\theta,\alpha)} \mathbb{E}_{(s,a)\sim\mathcal{D}}\left[\mathbb{E}_{\omega\sim q(\omega)}\left[Q_{\psi_1}(s, \pi_\omega(s))\right] + \mathcal{L}_{\text{RL-ELBO}}(\theta, \alpha)\right], \tag{11}$$

where $\mathcal{L}_{\text{RL-ELBO}}$ serves as the policy constraint term, preventing poor OOD action selection.

## 3.3 POLICY DISTILLATION

We can distill and optimize the policy's weights $(\theta, \alpha)$ by maximizing (11). However, if the influence of the KL term in $\mathcal{L}_{\text{RL-ELBO}}$ is too strong in the early stages, the policy may become excessively sparse too quickly, hindering performance improvements. To achieve a balanced trade-off between

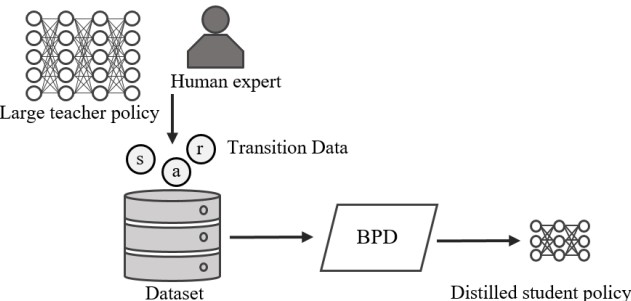

Figure 1: Schematic of the Bayesian policy distillation framework. From a pre-collected dataset, the control knowledge is distilled offline into a compact, small-sized neural network via the Bayesian policy distillation algorithm.

$Q$-value improvement and policy sparsity, and thereby promote stable learning, we propose the following practical loss function:

$$
\mathcal{L}_{\text{BPD}}(\theta, \alpha) = -\frac{1}{M} \Sigma_{m=1}^{M} \lambda Q_{\psi_1}(s_m, \pi_{\omega_m}(s_m))
$$
$$
+ \frac{|\mathcal{D}|}{M} \Sigma_{m=1}^{M} (\pi_{\omega_m}(s_m) - a_m)^2 + \eta \mathrm{D}_{\text{KL}}(q(\omega|\theta, \alpha)||p(\omega)). \tag{12}
$$

The loss function $\mathcal{L}_{\text{BPD}}$ (12) includes coefficients $\lambda$ and $\eta$, which control the contributions of the $Q$-value and KL divergence terms, respectively. By adjusting these coefficients, we can prevent the instability in learning that arises from rapid policy sparsification. In addition, the expectation over $\omega$ in (12) is approximated to a single sampled value of $\omega_m$ for each $m^{\text{th}}$ tuple. Such a sample-based approximation is useful in the RL framework, where the policy undergoes frequent updates. Furthermore, the intractable KL term can be approximated as follows (Molchanov et al., 2017):

$$
\mathrm{D}_{\text{KL}}(q(\omega_{ij}|\theta_{ij}, \alpha_{ij})||p(\omega_{ij})) \approx -k_1 \sigma(k_2 + k_3 \log \alpha_{ij}) + 0.5 \log(1 + \frac{1}{\alpha_{ij}}) + \text{const.}, \tag{13}
$$

where $k_1 = 0.63576, k_2 = 1.87320, k_3 = 1.48695$, and $\sigma$ is a sigmoid function. We use this approximation to calculate the KL term in $\mathcal{L}_{\text{BPD}}$. Lastly, to train the policy with random variables, additive noise reparameterization (Molchanov et al., 2017) is employed.

Considering all the above, we propose a learning algorithm for BPD. The student policy is trained by iteratively minimizing the objectives $\mathcal{L}_Q$ in (10) and $\mathcal{L}_{\text{BPD}}$ in (12). Next, we filter out weights where $\log \alpha_{ij} > C_{\text{Threshold}}$, considering them uninfluential. The filtered weights are set to zero, compressing the student policy network. As seen in Fig. 1, the static dataset $\mathcal{D}$ for computing $\mathcal{L}_{\text{BPD}}$ (12) can be constructed by observing human expert actions or leveraging a previously trained policy with a large-sized neural network.

### 3.4 Adjusting Weight Coefficients

Here, we discuss how to determine $\lambda$ and $\eta$ in the loss function $\mathcal{L}_{\text{BPD}}$ (12). Empirical experiments show that the excessive impact of KL-regularization in the early stage of training causes instability. Therefore, a coefficient $\eta$ is multiplied by the KL term. More specifically, $\eta$ is initially set to zero and annealed linearly during training, thereby reducing its impact in the early stages and gradually increasing its strength, contributing to the sparsity of the student policy through stable learning.

In addition, a weight coefficient $\lambda$ is employed to determine the weight to be given to the value-optimizing role of $\mathcal{L}_{\text{BPD}}$. Here, $\lambda$ is set to be proportional to $|\mathcal{D}|/\text{Average}(|Q_{\psi_1}(s, a)|)$, which is a similar normalizing technique introduced in (Fujimoto & Gu, 2021). This adaptive law is crucial as the $Q$-values and dataset size are highly task-dependent. If the dataset size is large, the impact of the BC-term in (12) will be substantial. In contrast, if the dataset size is small, the BC-term has a smaller weight coefficient according to the dataset size $\mathcal{D}$ and its effect is diminished. Thus, the inclusion of the weight coefficient $\lambda$ reflects the dataset size for more reliable $Q$-value updates, ensuring applying a unified objective $\mathcal{L}_{\text{BPD}}$ across various tasks.

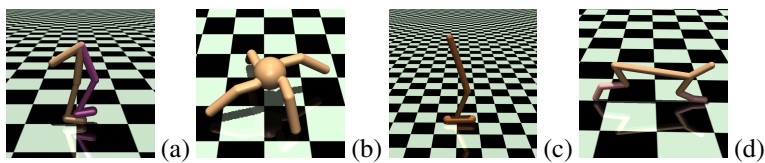

Figure 2: Multi-joint dynamics with contact continuous control environments: (a) Walker2d-v3, (b) Ant-v3, (c) Hopper-v3, (d) HalfCheetah-v3. The tasks aim to train a policy that encourages the robots to move forward quickly.

In a more formal representation, the coefficients $\eta$ and $\lambda$ can be written as follows:

$$\eta_n = \min(\frac{\nu}{N}n, 2) \tag{14}$$

$$\lambda = \frac{h \cdot |\mathcal{D}|}{\frac{1}{|M|}\Sigma_{(s_m, a_m) \in M}|Q(s_m, a_m)|}, \tag{15}$$

where $h$ is a proportional gain, $N$ denotes the total number of updates, $n$ is the current count of updates, $\nu$ is a hyperparameter for adjusting the annealing speed, and $M$ is a mini-batch set in the update stage. In the experiments, we set $N$ to $|\mathcal{D}|$ for all tasks. The details of BPD are summarized in Appendix A.

## 4 EXPERIMENTAL VALIDATION

We validate the proposed BPD through experiments with MuJoCo tasks from OpenAI Gym (Brockman et al., 2016), namely *HalfCheetah-v3, Walker2d-v3, Hopper-v3, Ant-v3* (Fig. 2). Two teacher policies were pretrained for each task: the expert-level policy with a high performance and medium-level policy with moderate performance. This setting is to show that a medium-level policy can be improved while being compressed.

The goal is to teach the robots to move forward quickly without falling. Hence, the moving speed at each step is reflected in the immediate reward. The experiments illustrate how effectively BPD can compress the student's policy network while preserving or enhancing the teacher's performance. To quantify the network compression ability, we define a measure called *sparsity* as follows:

$$\text{Sparsity } (\%) = 100 \cdot \frac{|\omega_s \neq 0|}{|\omega_t|},$$

where $|\omega_s \neq 0|$ is the number of non-zero weights in the student policy and $|\omega_t|$ is the total number of weights in the teacher policy. The teacher policy was trained with the soft-actor critic framework (Haarnoja et al., 2018) with two hidden layers consisting of 400 units and 300 units, denoted as (400, 300), which is one of the widely used common sizes in RL for MuJoCo tasks (Fujimoto et al., 2018; Haarnoja et al., 2018). Under these conditions, we could train expert and general-level teacher policies, from which two corresponding static datasets were constructed with one million transition tuples.

The proposed BPD was compared with well-known network compression methods: deep compression (DC) (Han et al., 2016), sparse variational dropout (SVD) (Molchanov et al., 2017), and TD3+BC, a widely used offline RL algorithm (Fujimoto & Gu, 2021). Originally, the DC method employed pruning, quantization, and Hoffman coding for network compression; however, for a fair comparison, only the pruning step was performed, ignoring the performance degradation caused by quantization and Hoffman coding.

In DC, SVD, and BPD, the student policies have a hidden layer size of (128, 128) for an expert-level teacher policy and (64, 64) for a medium-level one; the distillation process attempts to increase their zero-weights to the maximum. In the case of TD3+BC, the hidden layer size of the student policies was chosen to be (32, 32) for all tasks except for HalfCheetah-v3 (40, 40). The size setting results in a compression level comparable to that of other baselines, making it easy to compare performance changes depending on compression levels.

| | Environment | Teacher | DC | SVD | TD3+BC | BPD (ours) |
|---|---|---|---|---|---|---|
| Return | Ant (Expert) | 5364±1773 | 5033±388 | **5598±131** | 1857±598 | **5455±201** |
| | Ant (Medium) | 2642±493 | 2066±187 | 2309±97 | **3562±109** | **3136±81** |
| | Walker2d (Expert) | 5357±21 | 4244±686 | **4937±213** | 4877±364 | 4817±545 |
| | Walker2d (Medium) | 2256±1290 | 2815±350 | 3180±229 | **3737±44** | **3626±121** |
| | Hopper (Expert) | 3583±13 | 2891±365 | **3565±3** | 2223±1019 | **3134±549** |
| | Hopper (Medium) | 2066±1295 | 1384±239 | 2108±303 | **2861±307** | **3029±130** |
| | HalfCheetah (Expert) | 11432±90 | 9884±754 | **10677±217** | 7973±1430 | **10355±318** |
| | HalfCheetah (Medium) | 5938±53 | 5737±43 | 5844±19 | **5950±38** | **5850±45** |
| Sparsity | Ant-v3 (Expert) | N/A | 14.20% | 2.23% | 2.93% | 2.40% |
| | Ant (Medium) | N/A | 8.05% | 1.91% | 2.93% | 1.92% |
| | Walker2d (Expert) | N/A | 27.19% | 2.00% | 1.42% | 1.68% |
| | Walker2d (Medium) | N/A | 30.78% | 2.21% | 1.42% | 1.94% |
| | Hopper (Expert) | N/A | 22.92% | 1.70% | 1.22% | 1.35% |
| | Hopper (Medium) | N/A | 23.73% | 1.85% | 1.22% | 1.62% |
| | HalfCheetah (Expert) | N/A | 52.43% | 2.23% | 2.02% | 2.21% |
| | HalfCheetah (Medium) | N/A | 12.76% | 1.63% | 2.02% | 1.38% |

Table 1: Performance (return) and sparsity benchmark for several continuous control tasks. The figures in bold highlight the best and second-best performances across baselines.

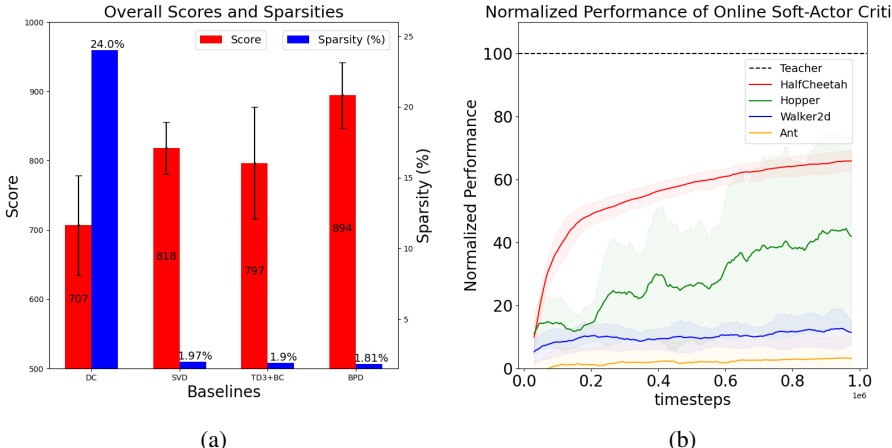

(a)                                                    (b)

Figure 3: **(a)** Overall scores and sparsities across the baselines. The overall score is calculated by summing up the normalized performance of all tasks, and the overall sparsity is obtained by averaging the sparsity of all tasks. **(b)** Normalized performances of small-sized policies trained with *online* soft-actor critic. The results show that training with a naively and overly compressed policy reduces sample efficiency or fails to enhance performance.

To evaluate network sparsity and performance, all algorithms were run with ten different random seeds on each task. The student policies were trained for one million time steps, and their performance was evaluated every 5,000 time steps. For each seed, the final performance was determined by averaging the last ten measured performances. The results from ten random seeds are averaged and summarized in Table 4.

### 4.1 IMPLEMENTATION DETAILS

For the proposed BPD, a nonlinear rectified linear unit activation function was adopted between hidden layers. The action space was constrained as $[0, 1]$ to ensure appropriate action selection for given environments. Furthermore, we consistently set $C_{\text{Threshold}}$ to 2 and $\nu$ to 4 for all experimental tasks. Since $C_{\text{Threshold}}$ and $\nu$ are consistently set without careful selection, room for improvement exists by choosing values suitable for each task. The hyperparameters adopted are listed in Appendix A.

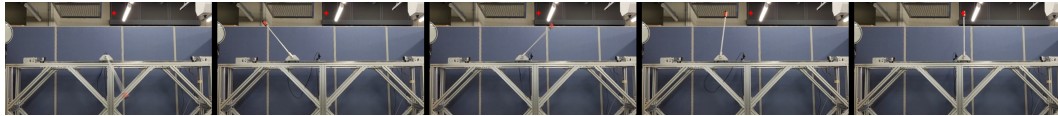

Figure 4: The real inverted pendulum system. The goal of the task is to swing up and balance the pole. The policy distilled through BPD successfully completed the task using only 1.5% of the original total number of parameters.

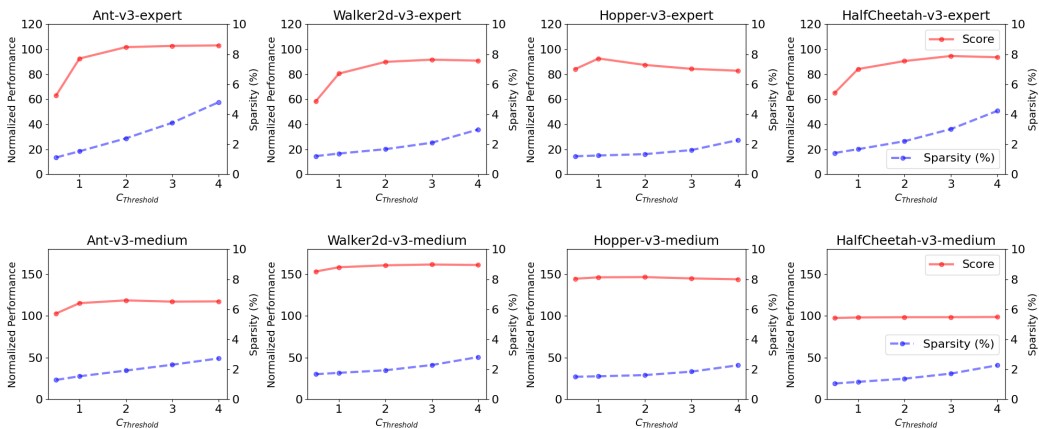

Figure 5: Normalized performance and sparsity comparison for different values of the thresholds: $C_{\text{Threshold}} \in \{0.5, 1, 2, 3, 4\}$.

### 4.2 SIMULATION RESULTS

The proposed BPD showed remarkable compression capability (approximately 1.8%) for expert- and medium-level teacher policies. The compressed student policies demonstrated minimal performance degradation and, in some cases, even improved performance. Table 4 shows that BPD outperforms other baselines at similar sparsity levels and has performance scores in the top two for almost all tasks.

We also compared the *overall* scores and sparsities of all the baselines. The performance scores were normalized to the teacher's $(= 100 \cdot \mathbb{E}[\text{Student Performance}]/\mathbb{E}[\text{Teacher Performance}])$ for each task and then averaged across all ones. The averaged overall performance scores and sparsities are illustrated in Fig. 3(a). Notably, DC showed significantly low compression capability compared to other baselines. This suggests that commonly used deep learning compression techniques may not be effective for BC in reinforcement learning despite high performance in tasks such as image classification.

### 4.3 ABLATION STUDY

BPD filters out uninfluential weights based on a prescribed $C_{\text{Threshold}}$ value. Therefore, we explored how variations in the $C_{\text{Threshold}} \in \{0.5, 1, 2, 3, 4\}$ affect the performance of the agent and the sparsity of the neural network. When $C_{\text{Threshold}}$ is small, e.g., 0.5, excessive network compression occurs, considerably degrading network performance. However, increasing the $C_{\text{Threshold}}$ did not result in a proportional improvement in performance, as shown in Fig. 5. This implies that once a certain network connectivity level is established, additional connections will not necessarily contribute to performance improvement.

Furthermore, another experiment was conducted to investigate whether a student policy with a small-sized neural network could achieve expert-level performance without a teacher one. Here, small-sized student policies were trained with a soft-actor critic algorithm under an *online* RL framework.

| Metric | Teacher Policy | Distilled Policy |
|---|---|---|
| Mean Normalized Score (10 trials) | 100.0 | 100.3 |
| Mean Inference Time (ms) | 1.36 | 0.30 |
| # of parameters | 72705 | 1108 |
| Memory (KB) | 290.82 | 4.43 |

Table 2: Policy distillation results in the real inverted pendulum task.

The hidden layer size of the student policies was chosen to be (32, 32) for all tasks except for HalfCheetah-v3 (40, 40). As shown in Fig. 3(b), the resulting performance does not reach an expert level, indicating that naively and overly compressed policy significantly lowers the sample efficiency or fails to enhance the performance.

## 4.4 REAL EXPERIMENT

For real-system validation, BPD was applied to an inverted pendulum system, where the goal is to swing and balance the pendulum (see Fig. 4). The state variable is constructed by stacking five consecutive observations, which includes position, velocity, $\sin \theta$, $\cos \theta$, and $\dot{\theta}$, where $\theta$ is the angle between cart and pole (Fig. 6 in App. B). Thus, the total dimension of the state space becomes 25 (5×5 observation). The action generated by the policy determines the reference velocity for the motor actuator. The teacher policy was pretrained using the real inverted pendulum system and DDPG algorithm (Lillicrap et al., 2016) with a hidden layer size of (256, 256). Then, we collected 300,000 transition data from the teacher, which took approximately 2.5 h. The teacher policy was then distilled through the proposed BPD to obtain a lighter and faster student policy. Sparse hidden layers in the student policy enabled efficient sparse matrix-vector operations, reducing inference time by representing the matrices in compressed sparse row format.

The teacher and distilled student policies are compared in terms of performance score, inference time, number of parameters, and memory storage size. The inference time was measured on an ARMV8-based processor, utilizing the SCIPY library that is one of PYTHON's computing algorithm libraries. The inference time was averaged over a total of 10,000 runs. The overall results are listed in Table 2. The inference time of the distilled student policy is 4.5 times faster than that of the teacher policy. Such a difference in the inference speed could be dramatically bigger on cheaper and highly resource-constrained devices that do not support Single Instruction Multiple Data (SIMD) or parallel programming. The required memory size was measured based on the number of non-zero weight parameters. The distilled policy uses only approximately 1.5% (4.43 / 290.82) of memory storage compared with the teacher policy. However, despite the significant memory savings, no notable performance degradation is observed but rather a slight performance improvement.

## 5 CONCLUSION AND LIMITATIONS

Deep reinforcement learning is becoming increasingly important in industries such as robotics, where practical applications require models to run on affordable, energy-efficient devices with limited computational resources. To meet this need, we introduce an efficient offline policy compression method called Bayesian Policy Distillation (BPD), which retrains a compact student policy network from a larger teacher network in an offline reinforcement learning setting. Our results demonstrate that BPD successfully reduces the size of the teacher policy network by 1%-2%, while achieving minimal performance loss. Notably, in some environments, we were able to achieve even higher performance than the teacher. Moreover, in a real inverted pendulum system, we confirmed that BPD can dramatically increase inference speed on devices with limited computational resources.

A drawback of BPD is its lack of real-time adaptability due to its offline nature. However, it is highly effective in situations where ongoing system interaction is too risky or costly. Additionally, when no pre-existing dataset is available, a teacher policy must first be trained, and expert data collected, which may lead to some sample inefficiency. That said, when a dataset exists, BPD allows for reuse, making it a cost-effective solution. Overall, we believe BPD offers a promising approach for advancing deep reinforcement learning across various industries, given the benefits of offline RL, its impressive compression performance, and significant improvements in inference speed.

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

# A  DETAILS OF BPD

## A.1  PSEUDO CODE OF BPD

---

**Algorithm 1** Bayesian Policy Distillation

---

1: **initialize** a static dataset $\mathcal{D}$, variational distribution $q(\omega|\theta, \alpha)$, policy $\pi_\omega(a|s)$, critics $Q_{\psi_{i \in \{1,2\}}}(s, a)$, target critics $Q^{\text{target}}_{\bar{\psi}_{i \in \{1,2\}}}(s, a)$, learning rate for the critic $\zeta_Q$, learning rate for the policy $\zeta_\pi$, soft update ratio $\tau$, annealing speed parameter $\nu$, filter threshold $C_{\text{Threshold}}$, and total number of iterations $N$
2: **for** $n = 1$ to $N$ **do**
3: $\quad \eta_n = \min(\frac{\nu}{N}n, 2)$ $\hfill \triangleright$*Update the coefficient $\eta$*
4: $\quad M = \{(s_m, a_m, r_m, s'_m)\}_{m=1}^{|M|} \sim \mathcal{D}$
5: $\hfill \triangleright$*Randomly sample a mini-batch $M$ from the dataset $\mathcal{D}$*
6: $\quad a' \sim \pi_{\omega \sim q(\omega|\theta, \alpha)}(\cdot|s')$
7: $\quad y = R(s, a) + \gamma \min_{i \in \{1,2\}} Q^{\text{target}}_{\bar{\psi}_i}(s', a')$
8: $\quad \mathcal{L}_Q(\psi_i) = \mathbb{E}_{(s,a,s') \sim M}\left[\left(Q_{\psi_i}(s, a) - y\right)^2\right]$
9: $\quad$ **for** $i = 1$ to $2$ **do**
10: $\quad\quad \psi_i^{\text{new}} \leftarrow \psi_i - \zeta_Q \nabla_\psi \mathcal{L}_Q(\psi_i)$
11: $\hfill \triangleright$*Update the critic*
12: $\quad$ **end for**
13: $\quad$ **if** $n \bmod$ policy update frequency $== 0$ **then**
14: $\quad\quad \theta^{\text{new}} \leftarrow \theta - \zeta_\pi \nabla_\theta \mathcal{L}_{BPD}(\theta, \alpha)$
15: $\quad\quad \alpha^{\text{new}} \leftarrow \alpha - \zeta_\pi \nabla_\alpha \mathcal{L}_{BPD}(\theta, \alpha)$
16: $\hfill \triangleright$*Update the policy*
17: $\quad$ **end if**
18: $\quad \psi_{i \in \{1,2\}} \leftarrow \psi_{i \in \{1,2\}}^{\text{new}}$
19: $\quad \theta \leftarrow \theta^{\text{new}}$
20: $\quad \alpha \leftarrow \alpha^{\text{new}}$
21: $\quad \bar{\psi}_{i \in \{1,2\}} \leftarrow \tau\psi_{i \in \{1,2\}} + (1 - \tau)\bar{\psi}_{i \in \{1,2\}}$
22: $\hfill \triangleright$*Update the parameters*
23: **end for**
24: Set weights $\omega$ zero where $\log \alpha > C_{Threshold}$ $\hfill \triangleright$*Sparsify the policy*
25: **Return** the sparsified policy $\pi$

---

## A.2  HYPERPARAMETER DETAILS

| Hyperparameters | Values |
|---|---|
| Policy update frequency | 2 |
| Total number of updates | 1 million |
| Static dataset size | 1 million |
| Mini-batch size | 256 |
| Optimizer | Adam |
| $C_{\text{Threshold}}$ | 2 |
| $\zeta_Q, \zeta_\pi$ | 0.0003 |
| $\gamma$ | 0.99 |
| $\nu$ | 4 |
| $h$ | 0.5 |
| $\tau$ | 0.005 |

Table 3: Hyperparameters for Bayesian Policy Distillation.

## B    REWARD DETAILS OF THE REAL EXPERIMENT

In this section, we provide details of the reward function used in the real inverted pendulum task. Let $\theta$ be the angle between the cart and pole, $\dot{\theta}$ be the angular velocity, $||a||^2$ be the norm of the action, and $p_x$ be the cart's position with respect to the horizontal axis. Then, the reward $r$ in the real-world inverted pendulum task is determined as:

$$r = r_\theta \cdot r_{\dot{\theta}} \cdot r_{\text{pos.}} \cdot r_{\text{act.}},$$

where

$$r_\theta = \frac{1 + \cos\theta}{2},$$

$$r_{\dot{\theta}} = \frac{1 + \exp\left(-\dot{\theta}^2 \cdot \frac{\log 10}{25}\right)}{2},$$

$$r_{\text{pos.}} = \frac{1 + \exp\left(-p_x^2 \cdot \frac{\log 10}{4}\right)}{2},$$

$$r_{\text{act.}} = \frac{4 + \max\left(||a||^2, 0\right)}{5}.$$

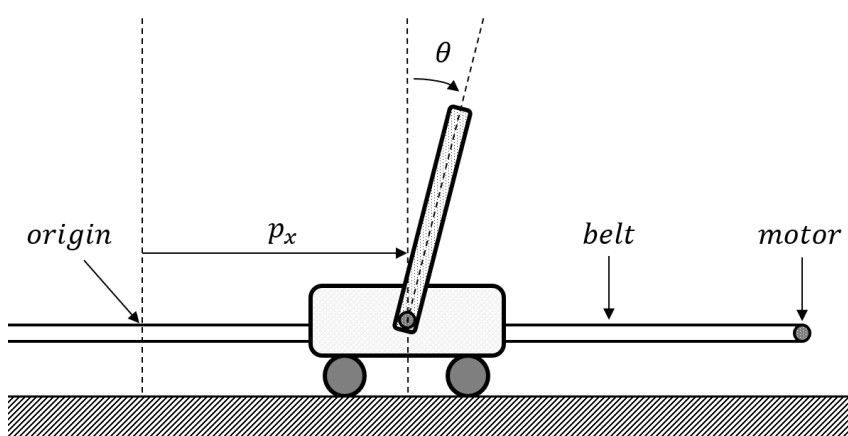

Figure 6: Schema of the real inverted pendulum system.

