# OpenReview forum: "Bayesian Policy Distillation via Offline RL for Lightweight and Fast Inference"
_ICLR.cc/2025/Conference — ICLR 2025 Conference Withdrawn Submission_

### Official Review · Reviewer_SMjd · 2024-11-02

**Soundness:** 4
**Presentation:** 2
**Contribution:** 3
**Rating:** 6
**Confidence:** 3

**Summary:**

This paper addresses the task of learning compressed DRL policies in an offline learning setting. The proposed method leverages a Bayesian training technique to simultaneously (1) mitigate the issues of offline training thanks to a behavior cloning regularization and (2) allow for a sparsification of the network thanks to a well chosen prior on the network weights.
The method is shown to perform favorably compared to methods coming from the supervised learning compression setting and a classic offline RL technique.

**Strengths:**

There are several things to like about this submission.
- First of all, I really like the addition of the proposed method as the modification of an already existing component. The paper leverages the presence of a behavior cloning loss in offline RL algorithms (such as TD3+BC), by using this loss to make the student network amenable to sparsification. This essentially extends the usual offline algorithms that incorporate a BC loss to be compressed simply by using a variational training approach for the student rather than a standard supervised learning one. I find this is a smart solution since it uses an element that was already there (the BC loss) and transforms it to obtain a new property (a controllable sparsity of the student).
- Figure 5 shows that the sparsification of the student network is very well behaved: as we lower the threshold value C, the sparsity (according to the definition of L358) monotonously “decreases” (except for Hopper-v3-medium) with the performance. This lets the experimentator decide what trade-off to strike between performance and sparsity.
- The paper includes a real-world experiment to demonstrate that the proposed method is indeed quicker to run than the complex teacher. This is particularly commendable as it demonstrates the good sparsity (through the inference speed) and performance of the proposed method in the concrete real-world setting that it is supposed to be tailored towards. Given that offline RL algorithms are particularly interesting for real-world applications, this is a very welcome level of evidence.
- Finally, I found the paper well structured (with, however, the notable absence of a related works section). The authors made an effort in slowly building their approach (perhaps even too much in Sec. 2!) and its rationale. Moreover, the text is well written. The equations are not overdone, and useful to illustrate the progressions of the text. The ideas were, overall, exposed very clearly.

**Weaknesses:**

At the same time, the paper could be improved on several aspects.

- First, I find that despite the clarity of most of the paper, the proposed approach was not sufficiently well placed in its environment. For instance, the paper uses a module usually intended to prevent instability due to OOD actions (the BC loss) and leverages it to a completely different effect (the compression of the policy). The idea is not problematic in itself (on the contrary), but I would have appreciated a discussion of the “twist” that this paper used regarding the BC loss compared to similar algorithms, maybe at the beginning of Section 3, to make the transition from the usual usage of the BC loss towards the new one. This aspect was exacerbated by the absence of an explicit related works section clearly comparing the proposed method and the existing ones, and a contributions section explicitly delineating the improvements brought by the BPD algorithm. Finally, unless I am mistaken, I find the method very close to TD3+BC, including the way to set the hyperparameters (Sec 3.4), obviously with the exception of the training and form of the student policy. If this is indeed the case, I think a clearer comparison with TD3+BC would be important to add. It should be made clear that TD3+BC is not intended for compression (TD3+BC), even though the naming of the metric “sparsity” (L358) can make this point confusing when looking at the results in Table 1.
- I find the concrete performance improvements of the proposed approach to be relatively modest. Namely, at a comparable sparsity level, BPD has a rather comparable performance to TD3+BC (Table 1). Without an inherent sparsity mechanism, a small network trained with TD3+BC leads to a policy with roughly as many active parameters as the one of BPD, for a marked performance loss in only 3 environments out of 8. Given that TD3+BC seems to be the method BPD is based off, where the number of parameters is controlled by the size of the student network rather than a sparsity inducing prior, I would have expected it to perform significantly worse than BPD (while in practice TD3+BC gets a higher mean score in half the environments).
- Regarding the reporting of the performance, selecting the 2 highest performances seems like an arbitrary choice. I would find it more fair to select the methods that perform the best, including the ones for which the confidence interval intersect. Alternatively, the reliable library [1,2] provides great tools to compare different algorithms (I do not expect the authors to re-illustrate the results with rliable since other works in offline RL have adopted this Table format, eg [3]).
- I found the naming of the “sparsity” metric (L358) pretty confusing since it involves both the teacher network and the student network, such that a method that leads to a dense network and without a sparsity mechanism (TD3+BC) leads to a network with a very high “sparsity” metric. Maybe a naming scheme that makes it clear that the number is computer w.r.t the teacher (for instance, "teacher compression ratio" -- this is just a suggestion, others might be better) would be helpful.
- I found that unlike most of the paper, the paragraph at lines 42-62 was overall pretty confusing (especially L56-62). L49-51, online algorithms not sufficiently tuned are indicated to perform poorly because they interact with the environment: why is that a problem? I find the argument presented at L69 about offline training much more compelling. The meaning of the sentence at L56-57 was also vague to me. At L58-59: “aforementioned drawbacks”: I struggled to find the exact drawbacks that were referred to.

_Minor remarks_:
- Could you please indicate what the +/- mean exactly (what uncertainty measure) in Table 1?
- In Sec. 4, the student policies are chosen to be a smaller size than the teacher network. I am not sure why this is the case and how this size was chosen, since Sec 3.1 indicates that the student network size does not need to be set a priori (L226).
- I found Fig. 1 to not be very informative. As a personal opinion, I would have found a diagram with the different elements in the loss and their effects more useful.
- L374-375: “increase their zero-weights to the maximum”: I did not understand this sentence, could you please rephrase it?
- In Sec 4.4, you could precise that Appendix B contains the information about the computation of the score.
- L55: missing dash in Kullback-Leibler
- L464: Table 4 does not exist, you likely meant Table 1?


[1] https://agarwl.github.io/rliable/,
[2] Agarwal, Rishabh, et al. "Deep reinforcement learning at the edge of the statistical precipice." Advances in neural information processing systems 34 (2021): 29304-29320.
[3] Yang, Rui, et al. "Towards robust offline reinforcement learning under diverse data corruption." arXiv preprint arXiv:2310.12955 (2023).

**Questions:**

- Can you make the same ablation study as Fig. 5 with increasing network size for TD3+BC? How do the curves compare?
- How should you choose the size of the policy network trained with BPD?
- In Sec 4.4, you compare a teacher policy with a distilled policy learned with BPD on a real world pendulum. Given that TD3+BC (using a small network) was competitive with BPD in several environments, I am really curious to know how a small network trained with TD3+BC (at a similar sparsity level to BPD) would perform in this real-world inverted pendulum environment, and I think such a result would be helpful in demonstrating further the benefits of BPD compared to TD3+BC. Would it be possible to perform this experiment?
- You set C to 2 (L429) while indicating that the literature “widely use[s]” 3 (L183). How did you decide on this number?
- TD3+BC (with a small policy network) seems to be a very strong baseline (according to Table 1). Do you anticipate instances where BPD would be much more performant than TD3+BC?

---

### Official Review · Reviewer_fsrq · 2024-11-03

**Soundness:** 2
**Presentation:** 2
**Contribution:** 2
**Rating:** 3
**Confidence:** 4

**Summary:**

The paper introduces a Bayesian Policy Distillation (BPD) offline policy compression method that retrains a compact student policy network from a larger teacher network. Experimental results reveal that the proposed BPD successfully compresses the policy networks, making them lighter and achieving faster inference time.

**Strengths:**

1. The proposed method creates a lightweight and sparse offline policy suitable for situations with limited computational performance and expensive data collection costs.
2. Experiments were conducted in both simulated and real environments.

**Weaknesses:**

1. The framework diagram in Figure 1 is too simplistic. It cannot explain how BPD is applied to the learning of the student network.
2. The symbol "p" denotes different meanings in this paper. In Section 2.1, "p" represents the transition probability function, while in Section 2.2, "p" represents the posterior.
3. In Table 1, the names "Ant-v3" and "Ant" are not consistent.
4. From the experimental results presented in Table 1, the Return and Sparsity are not optimal in most tasks.
5. Sections 4.1 and 4.2 refer to "Table 4," but there is no "Table 4" in the paper; please check if it should be "Table 1."
6. The second experiment in Section 4.3 lacks persuasiveness for the paper's argument, as the student network compresses the model by learning from the teacher network, while the experiment learns a small network without any teacher guidance. This does not provide sufficient evidence for performance degradation in the student network's compression. Comparisons should be made between different sizes of student networks under the guidance of a teacher network to illustrate this.
7. The order of Figure 4 and Figure 5 should be consistent with the order in which they are referenced in the paper.

**Questions:**

1. In equation (10), is R(s,a) obtained from interacting with the environment? As far as I know, in offline RL, the reward should come from offline data.
2. In the experiments of Table 1, why is the student network size set separately for the TD3+BC method? Additionally, is it reasonable to have the same network size for the Expert and Medium settings in this method? Does the significantly lower Return of the Expert in the Ant task result from this setting?

---

### Official Review · Reviewer_AKQ5 · 2024-11-05

**Soundness:** 3
**Presentation:** 2
**Contribution:** 2
**Rating:** 3
**Confidence:** 3

**Summary:**

The authors present a method for batch constrained offline RL, from a "distillation" perspective.

**Strengths:**

- The introduction is written very well
- The experiments are carried out in a rigorous fashion, considering different environments and repetitions.
- Section 2 introduces the relevant concepts for the following derivations

**Weaknesses:**

**Significance:**
1. I find the “distillation” perspective here not very fitting. Essentially you are doing batch constrained RL. Because next to the behavior cloning you are also doing optimization (Eq. (10,11)). From this it follows that the  assumption is that the behavior policy (your teacher) is not optimal.
So the behavior policy may or may not perform very well. Then you optimize a policy, and add a constraint that it should not be too different from the behavior policy (because of safety, or risk of OOD). You now make this optimized policy sparse to achieve greater robustness (on top of the Bayesian modeling approach). The distillation approach only conceptually makes sense if the behavior policy is already good or near-optimal—otherwise, why distill imperfect information? But in that case, one would not need to do Eq. (10,11) and could just approximate the behavior policy directly using Eq. (8-9).

2.  Usually the biggest memory/scale requirements (and thus need to distill things) in RL do not come from the actor (the policy) but the critic (the Q-value in your case).   Modeling  "what to do at which state" (policy) is usually not scale-bounded compared to modeling "why to do what at which state" (Q/Value function, transition model).

**Clarity & Writing:**
The writing in Section 3 needs significant improvement and lacks clarity.

**3.1**


1. The term “student policy is very confusing in this context and was not introduced. In Eq. (8) the policy is  once referred to as “student policy” (“Then, the student policy is trained by solving the following minimization problem”)  and then as “target policy” (“where \pi_w is the target policy parameterized by w”).

So \pi_w is both a “student” as well as  the “target” policy. Then the authors say: “and is trained using BC. However, when the student policy size is excessively small, maintaining the performance of the teacher policy becomes challenging (Rusu et al., 2015).”  Now there is also a “teacher” policy, which does not follow at all from BC introduction earlier. Because \mathcal{D is the dataset, the initial batch, generated by “the behavior policy” and  the minimization of Eq. (8) is referred to as the student policy, I can only assume that the behavior policy is the “teacher”.

2. In Eq. (9) you are doing a ELBO minimization of (8) using a VI. Then you say “L_{RL-Elbo) in (9) includes a term for BC as in (8)”. This is again very confusing.  EQ. (9) IS the BC term when modeled in a Baysian/VI way. The KL term follows from the Bayesian modeling perspective.

**3.2**
*“Therefore, learning a general behavior that makes good actions for states not included in the static dataset is considered when training the Q-function.”*

A behavior policy is not learned or optimized in offline RL, as it is fixed due to generating the batch. Its specifics (and whether it can even be approximated) are unknown. While one can approximate the behavior policy, in my understanding, that is not your approach. You are performing batch-constrained optimization to learn a policy (Eq. (10-11) that also remains close to the data.

*““a’ can be viewed as a combination of the action from the deterministic policy \pi_w and a random perturbation, where \overline{\pi} ) represents the mean value.”*

This is misleading. You are using a Bayesian neural network (BNN)-based approach. The distribution over the actions originates from the epistemic uncertainty over the parameters, not from a “stochastic policy” or “random perturbation.” You sample from a posterior over network parameters, representing the belief about which policy is correct.

**Questions:**

n/a

---

### Note · Authors · 2024-11-13

**Comment:**

On behalf of my co-authors, I sincerely thank the reviewers for their thoughtful feedback and would like to withdraw our paper to avoid wasting their valuable time.

**Withdrawal Confirmation:**

I have read and agree with the venue's withdrawal policy on behalf of myself and my co-authors.